# Utilisation of Post-Activation Performance Enhancement in Elderly Adults

**DOI:** 10.3390/jcm10112483

**Published:** 2021-06-04

**Authors:** Michał Krzysztofik

**Affiliations:** Institute of Sport Sciences, Jerzy Kukuczka Academy of Physical Education in Katowice, 40-065 Katowice, Poland; m.krzysztofik@awf.katowice.pl

**Keywords:** older adults, physical activity, sarcopenia, mitigate ageing, complex training, health

## Abstract

With age, many physiological changes occur in the human body, leading to a decline in biological functions, and those related to the locomotor system are some of the most visible. Hence, there is a particular need to provide simple and safe exercises for the comprehensive development of physical fitness among elderly adults. The latest recommendations for the elderly suggest that the main goal of training should be to increase muscle power. The post-activation performance enhancement effect underpinning complex training might be an approach that will allow for the development of both muscle strength and velocity of movement, which will result in an increase in muscle power and improve the ability to perform daily activities and decrease injury risk. This article briefly introduces a complex training model adapted to the elderly with its potential benefits and proposes a direction for further studies.

## 1. Introduction

With age, many physiological changes occur in the human body, leading to a decline in biological functions, and those related to the locomotor system are some of the most visible [1]. The consequence of these phenomena is a reduced quality of life [2], an increased risk of falls and fractures [3], as well as the development of cardiovascular and respiratory diseases, osteoporosis or type 2 diabetes, and obesity [4,5]. Increasing overall physical activity and undertaking resistance training in combination with aerobic training has been recognised as a key component of an effective strategy to improve health and mitigate the consequences of the ageing process [6]. Age-related decline in performance is associated with alterations in the nervous and muscular systems [7]. As a consequence of the loss of muscle mass, strength, and changes in muscle architecture, the force–velocity relationship of muscles alters with ageing [8]. Those changes affect the daily function of the elderly in terms of strength applied and velocity of movement. Therefore, the latest recommendations for the elderly suggest that the main goal of training should be to increase muscle power and, specifically, the velocity component under varying load [6,8,9]. Power is the product of force and velocity; therefore, improving muscle power requires changing either the force component or the velocity component (or both) to cause this change. Thus, the ideal would be an approach that develops the entire force–velocity spectrum, the capacity of muscle to generate more force at all contraction velocities, as well as the maximum velocity contraction. Hence, it would be appropriate to introduce both high- and low-loaded exercises performed in an explosive manner (i.e., concentric phase of movements performed as quickly as possible) will increase the velocity component of power, which is critical for safety. Nevertheless, in real-life scenario, it is more likely an older adult will have to move a lower limb quickly to keep balance and avoid falling or react quickly when driving a car than to move a heavy object. 

Despite the increasing amount of research, there is still an opinion in society that older adults are too weak or fragile to undertake high-velocity and high-intensity training. Moreover, the latest research shows that these exercises are not dangerous, but it is a recommended form of physical activity to improve the quality of life and a number of physiological functions [6,10]. Therefore, there is still a need to educate, motivate, and increase the self-confidence and adherence of older people to undertake this form of physical activity, and it seems that this can be achieved by providing simple and safe group activities. Hence, there is a particular need to provide uncomplicated and highly effective solutions that will allow for the development of both, muscle strength and velocity of movement, which will result in an increase in muscle power and improve the ability to perform daily activities. Such a solution may be an approach based on the use of the post-activation performance enhancement (PAPE).

## 2. Post-Activation Performance Enhancement and Complex Training

A muscular phenomenon known under the term post-activation potentiation (PAP), which enhances muscular performances as a result of the recent voluntary contractile history [11,12], has become the focus of many strength and conditioning training programs. The primary mechanism responsible for PAP is the phosphorylation of the myosin regulatory light chain that leads to an increase in calcium sensitivity of the actomyosin complex [11,13]. Except that, the occurrence of performance enhancement may be also related to the other linked mechanisms, such as an increase in muscle temperature, fibre water content, and muscle activation which are assigned to the alternative term PAPE [13,14]. Therefore, the role of PAP on sports performance has been widely debated, and it has been suggested that when a performance-related PAP approach is used instead of a mechanistic one, the term PAPE should be used [13,14]. In the case of training practice, increased force and power output production are induced by a potentiation complex consisting of a conditioning activity (CA) (i.e., bench press), followed by an explosive activity with a similar movement structure (i.e., medicine ball chest throw) [12,15]. This approach is called complex training. Since generating high values of muscle power is a critical variable for athletes in many sports, it seems that with caution and common sense, some inspiration from these solutions when training the elderly can be drawn. It is clear that the training methods used in sport are often extremely demanding and require adaptation and simplification to enable their implementation by older people, which is not impossible. 

Nowadays, complex training is considered an effective method of integrating the development of muscle strength and power [16]. The adaptive processes following this method of training have been attributed to the long-term translation of acute enhancement in muscle contractility properties induced by CA. Moreover, the long-term performance improvement following progressively performed complex training may relate to several neural and morphological factors [17]. It seems that explosive exercises, such as ballistic actions (i.e., post-CA exercises) may promote the recruitment of larger motor units, enhance their firing frequency and synchronisation, thus raising the potential for positive strength and power adaptation to occur [17,18]. Furthermore, high-loaded resistance exercises (i.e., CA) may also increase motor unit synchronisation and force production as well as increase muscle mass [19]. Therefore, it seems that complex training combines the benefits of both, explosive and high-loaded exercises.

The effects of complex training have been supported by extensive research assessing its acute and chronic impact on maximum strength, muscle power, sprint, and sports performance adaptations, also in comparison with other training methods [16,17,18,19,20]. Nevertheless, some studies did not show superior effects of that method to plyometric or traditional resistance training [20,21]. Moreover, the magnitude of PAP slightly decreases with ageing [22,23,24,25]. A study by Hicks et al. [23] showed that twitch potentiation was significantly greater in young than in elderly adults (241% vs. 166%). Interestingly, a later study showed that age-related differences in PAP occurred immediately after conditioning MVC and disappeared after 1 min [25]. This reduced PAP capacity in elderly adults is likely associated with a change in excitation–contraction coupling mechanisms [26] and factors related to muscle fibre atrophy [27,28]. In addition, a reduced myosin regulatory light chain phosphorylation during the CA has been observed in atrophied muscles and may have contributed to the diminished PAP in elderly individuals [29,30]. Even though the PAP response is reduced, it still occurs, which means that the elderly can benefit from properly designed training. For example, Hicks et al. [23] showed that the magnitude of potentiation significantly increased after 12 weeks of resistance training in the elderly (166% vs. 192%).

Appropriate selection of CA modalities as well as those performed post activation may induce less stress on the cardiovascular system and decreased perceived exertion, compared to traditional resistance and plyometric exercise performed separately [31]. In addition, this approach allows gaining the benefits from both, traditional resistance and plyometric training, in a single session, thus providing a comprehensive development across the entire force–velocity spectrum. Further, it is possible to exercise with greater frequency (high-velocity and high-resistance exercises in a single session) and lower volume (might be spread out on several sessions). Moreover, the exercise sessions might be more enjoyable by increasing the variety of exercises.

### 2.1. Conditioning Activity Modalities

To date, evidence showing that the PAPE effect can be effectively elicited in several ways, by high-intensity resistance exercises (>80% one-repetition maximum; 1RM) [12], moderate-intensity resistance exercises (40–60%1RM) with barbell velocity loss control [32], eccentric-overload resistance exercises [33,34,35,36], as well as plyometric exercises [37]. The most significant factor of achieving performance enhancement is that potentiation induced by the CA has to exceed the fatigue produced at the same time [38]. It seems that for the elderly, the most appropriate CA solution in terms of safety and efficacy would be the application of chosen eccentric training methods [39,40]. A slow tempo of movement and flywheel inertial method has been supported as an efficient strategy for muscle strengthening and increased functional capacity in the elderly [41,42,43]. Moreover, they were also efficient in inducing the PAPE effect [35,44]. A study by Wilk et al. [43] indicated that the PAPE effect is observed after a slow (6 s) eccentric phase of movement using 70%1RM. Furthermore, moderate-intensity inertia (three sets of six repetitions at 0.03 kg·m^2^) also has been shown as an effective CA to elicit the PAPE effect [37]. Moreover, it seems that the execution of exercises with the flywheel places lower technical requirements, in comparison to free-weight resistance exercise, while the execution of the eccentric phase with a longer duration may increase attention to proper positioning and form, and focus on muscle contraction during exercise. Additionally, eccentric contractions generate the highest forces with uniquely low energy costs, which constitute great characteristics for resistance training intervention to counteract the age-related muscle mass and strength [45]. 

### 2.2. Intra- and Inter-Complex Activity Modalities

The greatest improvement in performance is most often noted between 5–7 min after the CA for both lower- and upper-body exercises [16,46]. Consequently, the training density (number of repetitions per training unit) is low. The solution may be the approach proposed by Lim and Barley [47] consisting of the introduction of activity within the intra-(between CA and post-CA) and inter-complex (between sets of complex exercises) rest interval. These exercises should be of low intensity and should not be demanding due to the importance of the relationship between fatigue and potentiation in achieving the effect of performance enhancement. It seems that the mobility (e.g., serratus anterior wall slides) or stability drills (e.g., seated marching on stability ball) proposed by the Lim and Barley [47] for muscle groups not involved in the complex would be appropriate. Consequent to these exercises, the elderly will be able to improve muscle balance, posture, and overall body coordination, which will open up a number of new exercises they will be able to perform in later phases of training, without extending workout time. However, according to the author’s knowledge, no studies have been carried out so far to verify whether physical activity inside the complex will have a negative impact on the effect of performance improvement. Moreover, there are investigations showing higher effectiveness of active recovery involving the same muscles that were active during the CA in fatigue recovery than active exercise using the muscles that were not involved in the exercise [48]. Therefore, this should be approached with caution. The solution that would have the slightest disturbance to the occurrence of the PAPE effect may be respiratory training (breathing exercises, i.e., diaphragm breathing), which could potentially contribute to the improvement of cognitive and pulmonary functions [49]. Additionally, the implementation of simple cognitive tasks inside or between the complexes would not be demanding. Performing a simple and choice reaction time as well as working memory tasks (e.g., N-back task participant have to respond whenever a stimulus is presented, which is the same as the one showed *n* trials previously) could counteract a decrease in cognitive functions such as attention and memory. It seems interesting considering the positive relationship between physical activity and cognitive functions [50]. Nevertheless, there is a particular need for future research evaluating the effect of various exercises within and between the complex on the effect of subsequent performance improvement.

### 2.3. Post-Conditioning Activity Modalities

Most of the PAPE studies investigated the effects of a CA on a subsequent jump, vertical jump, sprint, throw, and upper-body plyometric and ballistic performances [12]. Plyometric exercises have long been used to increase reactive strength [51]. They are based on the use of the stretch-shortening cycle (SSC), in which the pre-activated muscle is first stretched and then shortened with a minimal transition delay. It was confirmed that contractions performed after a pre-stretch generate greater power output, compared with the simple shortening contractions. The reason for this phenomenon is related to the muscle–tendon unit, which is stretched and then rapidly shortened [52], in which the elastic properties of the muscle are involved in the storage and release of elastic energy. The use of plyometric exercises in an older population has previously been recommended as a potential preventive measure against age-related changes in muscle function [9]. The main problem with using such programmes is that plyometric exercises involving repetitive cycles of hard braking, followed almost immediately by the rapid acceleration in the opposite direction, appear difficult for older adults to perform without risk of injury. To circumvent this problem, recently proposed methods such as suspension training (e.g., assisted jumps) or trampoline-based plyometric exercise can be useful [10,53,54]. Exercises performed in a partial weight-bearing position (against the wall) manner can also be a good and easy-to-implement solution. Those approaches decrease impact forces upon landing; moreover, trampoline-based plyometric exercise might be performed seated (on a sled) with a defined trajectory or on the trampoline with supporting bars, reducing injury risk. 

Moreover, simple ballistic exercises could be considered as safe choices since the external load is projected onto the flight phase (e.g., ball throws), allowing to omit repetitive cycles of hard braking required during high-velocity non-ballistic and plyometric exercises. The advantage is that ballistic exercise was found to generate greater speed and power than the non-ballistic alternative [55,56]. The rationale for these increments is the need to accelerate during the concentric phase of movement to project the object onto the flight phase. Consequently, the acceleration path of the object is increased, requiring greater force to be applied to the object and higher speeds to be achieved, resulting in greater power output [57,58]. In addition, ballistic training increases the cross-sectional area of type IIx muscle fibres that are engaged in short-term explosive movements [59].

### 2.4. The Proposed Model of Complex Training for the Elderly

Taking into account the above-mentioned considerations and scientific reports, it seems that the properly designed complex training can constitute a holistic approach to counteracting the ageing process (Table 1). In an adapted model for the elderly, a moderate-intensity inertial training and/or slow eccentric phase of the movement as a CA may be appropriate to the initial phase of long-term complex training, with a gradual increase in movement speed, volume, and intensity in subsequent phases of periodisation (Table 2). Next, moderate-, and high-intensity machine-based or free-weight resistance exercise with velocity loss control (e.g., until mean barbell velocity dropped to 90% of that reached in the first repetition) could be incorporated as a CA. However, if velocity loss control is not possible due to the lack of access to needed equipment, then a combined rate of perceived exertion and repetitions in the reserve scale can be a good alternative [60]. 

The use of an active recovery within and between the complex will ensure greater training density and the possibility of introducing low-intensity corrective, stabilising, mobilising, and breathing exercises or cognitive tasks. This approach would allow for the development of additional components of fitness and cognitive functions simultaneously to mitigate the consequences of the ageing process within the same exercise session. However, there is no evidence as to whether these exercises will not affect the PAPE effect and therefore needs to be verified by future investigations.

While in the case of post-CA exercise, trampoline-based plyometric exercise (with the supporting bars or on the sledge) [53,61] can be considered as the safest option for the elderly, suitable for the first stage of periodisation in longitude complex training. Then, this could be progressed by the introduction of suspension exercises (e.g., jump squats using suspension straps). If it is not possible to introduce those solutions, exercise in a partial weight-bearing position might be a good choice (e.g., wall plyometric push-ups or wall sprints), followed by the ballistic exercises (e.g., jump squats). Additionally, for these exercises, the use of the combined rate of perceived exertion and repetitions in the reserve scale to adequately determine the training intensity and volume seems most appropriate with no equipment needed [60]. In regard to the previous studies on the periodisation of resistance training in the elderly and current recommendations, it seems that training sessions should be performed with an interval of approximately 48 h [6,62].

## 3. Conclusions, Practical Implication, and Directions for Further Studies

Although complex training has not been studied in the elderly, studies showed a beneficial effect of both high-intensity resistance and explosive training used separately. Therefore, it can be supposed that the mechanisms underlying its effectiveness in athletes could also be transferred to the elderly. The proposed model of complex training for the elderly may develop several components of fitness simultaneously to mitigate the consequences of the ageing process within the same exercise session. Nevertheless, the main limitation is that the proposed model is purely hypothetical, thus should be treated with caution. Furthermore, as, it is based on studies carried out on healthy elderly it may not be recommended for other older populations who have orthopaedic or other medical contraindications. Moreover, no study has yet investigated the effect of programming activity within the intra-complex and inter-complex recovery interval on subsequent performance improvement. Therefore, future studies have to verify the potential benefits of complex training after appropriate optimisation for the elderly to improve physical performance. Research protocols could compare the effects of complex training with traditional resistance training and other strength-oriented training methods, both in improving physical fitness and health, as well as increasing willingness and long-term adherence to physical activity. The proposed evidence-based model of complex training for the elderly may constitute a promising tool for exercise physiologists, strength and conditioning coaches, or personal trainers.

## Figures and Tables

**Table 1 jcm-10-02483-t001:** The proposed model of complex training adapted for the elderly.

**Phase 1 (Accumulation)**						
	**Exercise**	**Sets**	**Repetitions/Duration**	**Intensity/External Load**	**Tempo**	**Rest Intervals**
**Lower-Body Complex**						
Conditioning Activity	Barbell Squat	2–4	3 RIR	70%1RM	6/0/1/0	5–7 min **^^^**
Intra-Complex Activity	Simple Reaction Time	2–4	~3 min	-	-	
Post-Conditioning Activity	Wall Sprints	2–4	8–15 s	Body Mass	X/0/X/0	
Inter-Complex Activity *	‘Jug’ breathing ^#^	2–4	~2 min	Body Mass	-	4 min ^$^
**Upper-Body Complex**						
Conditioning Activity	Bench Press	2–4	3 RIR	70%1RM	6/0/1/0	5–7 min **^^^**
Intra-Complex Activity	N-Back Task	2–4	~4 min	-	-	
Post-Conditioning Activity	Wall Plyometric Push-Ups	2–4	8–12	Body Mass	X/0/X/0	
Inter-Complex Activity *	‘Semi-blocked’ breathing ^#^	2–4	~2 min	Body Mass	-	4 min ^$^
**Phase 2 (Intensification)**						
	**Exercise**	**Sets**	**Repetitions/Duration**	**Intensity/External Load**	**Tempo**	**Rest Intervals**
**Lower-Body Complex**						
Conditioning Activity	Barbell Squat	2–4	3 RIR/10% VLC	80%1RM	2/0/X/0	5–7 min **^^^**
Intra-Complex Activity *	Choice Reaction Time	2–4	~3 min	-	-	
Post-Conditioning Activity	Mini Trampoline Jumps	2–4	8–15 s	Body Mass	X/0/X/0	
Inter-Complex Activity *	Yoga Cat-Cow Pose	2–4	15	Body Mass	2/2/2/2	4 min ^$^
**Upper-Body Complex**						
Conditioning Activity	Bench Press	2–4	3 RIR/10% VLC	80%1RM	2/0/X/0	5–7 min **^^^**
Intra-Complex Activity *	N-Back Task	2–4	~4 min	-	-	
Post-Conditioning Activity	Medicine Ball Chest Pass	2–4	8–12	RPE 4	X/0/X/0	
Inter-Complex Activity *	Hip Thrust	2–4	10–15	Body Mass	2/2/2/2	4 min ^$^

* This exercise should be programmed with caution; #—description of the exercise presented in Ferreira et al. [49]; **^^^**—between conditioning activity and post-conditioning activity; ^$^—between complexes; RIR—repetitions in reserve; RPE—rate of perceived exertion; 1RM—one-repetition maximum; VLT—velocity loss control; 2/0/X/0—e.g., ‘2’ a 2 s eccentric phase, ‘0’ no rest in the transition phase, ‘X’ a concentric phase performed as fast as possible, and ‘0’ no rest before the next repetition.

**Table 2 jcm-10-02483-t002:** Example of periodisation model for complex training adapted for the elderly.

	Phase 1 (Accumulation)	Phase 2 (Intensification)
Duration of phase	3–6 weeks	3–6 weeks
Frequency/week	2–3 days	2–3 days
Repetitions or Duration	2–4 RIR */10 s ^#^	1–3 RIR */10 s ^#^
Complex Sets	2–4	2–4
Intensity	65–75%1RM */RPE 4 ^#^	75–85%1RM */RPE 5 ^#^

*—relates to conditioning activity; #—relates to post-conditioning activity; RIR—repetitions in reserve; RPE—rate of perceived exertion; 1RM—one-repetition maximum.

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
