# Peer review of "Utilisation of Post-Activation Performance Enhancement in Elderly Adults"

_jcm, 2021, doi:10.3390/jcm10112483_

Round 1

Reviewer 1 Report

Dear author,

I would like to congratulate you on the relevance of the present work and the interesting effort to carry out this study, in a clear attempt to bridge the gap between literature and application to decision making among exercise physiologists, strength and conditioning coaches or personal trainers. In my opinion the overall appreciation of the manuscript is highly positive, as the present work can provide an interesting, up to date and evidence-based model, particularly directed to a non-sporting population, such as elderly adults. I find the manuscript displayed in a generally well-written English, with an acceptable flow throughout the text, although it could be improved. Nevertheless, I have some concerns or comments for your consideration, particularly regarding the consistency in some raised aspects of the rationale, and the potential benefits and limitations, as following:

Abstract

Line 17: I suggest the author to change the total number of keywords (#eleven), since three to ten pertinent keywords need to be added, according to the instructions for authors.

  1. Introduction

The introduction section enables the potential reader with a proper framework, providing the sequential elements for the rationale, although the importance of a force-velocity training based among older adults could be highlighted.

Line 38: Please highlight the importance of muscular strength, particularly the morphological and neural factors, and especially, the force-velocity relationship applied to resistance training in elderly people [see Suchomel et al., (2018), Orssato et al. (2019) or Raj et al. (2010), for example. The final version of the manuscript would benefit from it.

  1. Post-Activation Performance Enhancement and Complex Training

Lines 52–56: This sentence should appear after Line 59 (“… complex training.”), in order to become more fluent.

Line 68: I find it a bit strange why literature regarding PAPE do not reflect about the neural factors in progressive resistance exercise – motor unit recruitment, rate coding, motor unit synchronization, and neuromuscular inhibition (Suchomel et al., 2018), for example. Could you provide any insight about this?

Lines 69–95: The extension of the second paragraph is to long. Please consider reorganize it, and also improve its fluency.

2.1. Conditioning Activity Modalities

I think the assumptions assumed from the information considered in this section need further explanations and support.

Lines 103–105: Could you please explain this statement, providing a proper support.

Lines 111–114: Part of this statement is quite similar to a previous one (Lines 87–90). Please change.

2.2. Intra-Complex Activity Modalities

I think the assumptions assumed from the information considered in this section need further consistency and clarity.

2.3. Post-Conditioning Activity Modalities

Line 147: I do not feel convinced with this affirmation. When looking at the cited paper (Faulkner et al., 2007), it briefly makes a reference to plyometric training, generalizing across all ages by citing the Wilson et al. (1993) study, which involves athletes sixty-four subjects with a mean age of 22.1±6.8 years.

2.4. The Proposed Model of Complex Training for the Elderly Adults

The proposed model, as presented in Table1, represents a practical method, based on a proper and pertinent rationale. Although some aspects underpinning this model of complex training adopted for the elderly adults, the hypothesis seem to be particularly suitable and constitutes a promising tool for exercise physiologists, strength and conditioning coaches or personal trainers.

Line 196: Do you mean “in Ferreira et al. (2015)2 instead of “in Fere”? If so, please correct it.

  1. Conclusions

Could the author provide further insights over the implications for the possible benefits of this? In addition, every study has its own limitations, and it should be clearly recognized the present study weaknesses and suggest further studies, besides recovery.

Lines 213–220: Not applicable to the present work. Please remove it.

References

Line 230: Title is in caps. Please correct.

Author Response

Dear author,

I would like to congratulate you on the relevance of the present work and the interesting effort to carry out this study, in a clear attempt to bridge the gap between literature and application to decision making among exercise physiologists, strength and conditioning coaches or personal trainers. In my opinion the overall appreciation of the manuscript is highly positive, as the present work can provide an interesting, up to date and evidence-based model, particularly directed to a non-sporting population, such as elderly adults. I find the manuscript displayed in a generally well-written English, with an acceptable flow throughout the text, although it could be improved. Nevertheless, I have some concerns or comments for your consideration, particularly regarding the consistency in some raised aspects of the rationale, and the potential benefits and limitations, as following:

Reply: I would like to thank the Reviewer for this favourable opinion and for appreciating my scientific effort. I’m grateful for the commitment and very valuable comments that helped me to improve the quality of the manuscript. I hope that the revised manuscript adequately addresses all the raised issues.

Abstract

Line 17: I suggest the author to change the total number of keywords (#eleven), since three to ten pertinent keywords need to be added, according to the instructions for authors.

Reply: Thank you for pointing this out. I have removed one of them, currently, there are ten words.

  1. Introduction

The introduction section enables the potential reader with a proper framework, providing the sequential elements for the rationale, although the importance of a force-velocity training based among older adults could be highlighted.

Line 38: Please highlight the importance of muscular strength, particularly the morphological and neural factors, and especially, the force-velocity relationship applied to resistance training in elderly people [see Suchomel et al., (2018), Orssato et al. (2019) or Raj et al. (2010), for example. The final version of the manuscript would benefit from it.

Reply: Thank you for this valuable comment. I have added the following sentences as suggested:

 “Age-related decline in performance is associated with alternations in the nervous and muscular systems [7]. As a consequence of the loss of muscle mass, strength, and changes in muscle architecture, the force-velocity relationship of muscles alters with ageing [8]. Those changes affect the daily function of the elderly in terms of strength applied and velocity of movement. Therefore, the latest recommendations for the elderly suggest that the main goal of training should be to increase muscle power, and specifically the velocity component under varying load [6,8,9]. Power is the product of force and velocity, so improving muscle power requires changing either the force component or the velocity component (or both) to cause this change. Thus, the ideal would be an approach that develops the entire force–velocity spectrum, the capacity of muscle to generate more force at all contraction velocities as well as the maximum velocity contraction. Hence, it would be appropriate to introduce both high- and low-loaded exercises performed in an explosive manner (i.e., concentric phase of movements performed as quickly as possible) will increase the velocity component of power, which is critical for safety.”

  1. Post-Activation Performance Enhancement and Complex Training

Lines 52–56: This sentence should appear after Line 59 (“… complex training.”), in order to become more fluent.

Reply: Thank you for this comment. I have reorganized this paragraph.

Line 68: I find it a bit strange why literature regarding PAPE do not reflect about the neural factors in progressive resistance exercise – motor unit recruitment, rate coding, motor unit synchronization, and neuromuscular inhibition (Suchomel et al., 2018), for example. Could you provide any insight about this?

Reply: Thank you for this excellent comment. I have added the following sentences: "The adaptive processes following this method of training have been attributed to the long-term translation of acute enhancement in muscle contractility properties induced by CA. Moreover, the long-term performance improvement following progressively per-formed complex training may relate to several neural and morphological factors [17]. It seems that explosive exercises, such as ballistic actions (i.e., post-CA exercises) may promote the recruitment of larger motor unit, enhance their firing frequency and synchronization, thus raising the potential for positive strength and power adaptation to occur [17,18]. Furthermore, high loaded resistance exercises (i.e., CA) may also increase motor unit synchronization and force production as well as increase muscle mass [19]. Therefore, it seems that complex training combines the benefits of both, explosive and high loaded exercises."

Lines 69–95: The extension of the second paragraph is to long. Please consider reorganize it, and also improve its fluency.

Reply: Done, a whole paragraph has been reorganized.

2.1. Conditioning Activity Modalities

I think the assumptions assumed from the information considered in this section need further explanations and support.

Lines 103–105: Could you please explain this statement, providing a proper support.

Reply: Thank you for this comment. I have added the following sentences for support: It seems that for the elderly the most appropriate CA solution in terms of safety and efficacy would be the application of chosen eccentric training methods [39,40]. A slow tempo of movement and flywheel inertial method has been supported as an efficient strategy for muscle strengthening and increased functional capacity in the elderly [41–43]. Moreover, they were also efficient in inducing the PAPE effect [35,44].

Lines 111–114: Part of this statement is quite similar to a previous one (Lines 87–90). Please change.

 Reply: Thank you for pointing this out. I have added the following sentences: “Besides, it seems that the execution of exercises with the flywheel places lower technical requirements in comparison to free-weight resistance exercise. While the execution of the eccentric phase with a longer duration may increase attention to proper positioning and form as well as focus on muscle contraction during exercise. Additionally, eccentric contractions generate the highest forces with uniquely low energy cost what constitute great characteristics for resistance training intervention to counteract the age-related muscle mass and strength [45].”

2.2. Intra-Complex Activity Modalities

I think the assumptions assumed from the information considered in this section need further consistency and clarity.

Reply: Thank you for this comment. I have added the following sentences for support and examples of the mobility and stability exercises in line 143-144)“Thanks to these exercises, the elderly will be able to improve muscle balance, posture and overall body coordination, which will open up a number of new exercises they will be able to do in later phases of training, without extending workout time.” and “Also, the implementation of simple cognitive tasks inside or between the complexes wouldn’t be demanding. Performing a simple and choice reaction time as well as working memory tasks (e.g., N-back task participant have to respond whenever a stimulus is presented that is the same as the one showed n trials previously) could counteract a decrease in cognitive functions such as attention and memory. It seems interesting considering the positive relationship between physical activity and cognitive functions [50]”

2.3. Post-Conditioning Activity Modalities

Line 147: I do not feel convinced with this affirmation. When looking at the cited paper (Faulkner et al., 2007), it briefly makes a reference to plyometric training, generalizing across all ages by citing the Wilson et al. (1993) study, which involves athletes sixty-four subjects with a mean age of 22.1±6.8 years.

Reply: You are right, many thanks for pointing this out. I have changed references to Vetrovsky et al., 2019

2.4. The Proposed Model of Complex Training for the Elderly Adults

The proposed model, as presented in Table1, represents a practical method, based on a proper and pertinent rationale. Although some aspects underpinning this model of complex training adopted for the elderly adults, the hypothesis seem to be particularly suitable and constitutes a promising tool for exercise physiologists, strength and conditioning coaches or personal trainers.

Reply: Thank you for this favourable opinion. I decided to add the inter-complex activity to effectively manage the rest between complexes. In addition, I have implemented the cognitive tasks as intra-complex activity as it might be a good, non-fatiguing choice to improve cognitive function.

Line 196: Do you mean “in Ferreira et al. (2015)2 instead of “in Fere”? If so, please correct it.

Reply: Thank you for pointing this out. Corrected.

  1. Conclusions

Could the author provide further insights over the implications for the possible benefits of this? In addition, every study has its own limitations, and it should be clearly recognized the present study weaknesses and suggest further studies, besides recovery.

Reply: Thank you for this relevant comment. Almost the whole paragraph has been changed: “Although complex training has not been studied in the elderly, studies showed a beneficial effect of both high-intensity resistance and explosive training used separately. Therefore, it can be supposed that the mechanisms underlying its effectiveness in athletes could also be transferred to the elderly. The proposed model of complex training for the elderly may develop several components of fitness simultaneously to mitigate the consequences of the ageing process within the same exercise session. Nevertheless, the main limitation is that the proposed model is purely hypothetical, thus should be treated with caution. Furthermore, as, it is based on studies carried out on healthy elderly it may not be recommended for other older populations who have orthopaedic or other medical contraindication. Moreover, no study has yet investigated the effect of programming activity within the intra-complex and inter-complex recovery interval on subsequent performance improvement. Therefore, future studies have to verify the potential benefits of complex training after appropriate optimization for the elderly to improve physical performance. Research protocols could compare the effects of complex training with traditional resistance training and other strength-oriented training methods, both in improving physical fitness and health, as well as increasing willingness and long-term adherence to physical activity. The proposed evidence-based model of complex training for the elderly may constitute a promising tool for exercise physiologists, strength and conditioning coaches or personal trainers.”

Lines 213–220: Not applicable to the present work. Please remove it.

Reply: Done.

References

Line 230: Title is in caps. Please correct.

Reply: Corrected, thank you.

Reviewer 2 Report

Manuscript Title: Utilization of post-activation performance enhancement in elderly adults

Manuscript Number: JCM1162818

General comments 

This manuscript provides a literary rationale for the use of complex training in the elderly. The text is well written and is easy to follow. Overall, in my opinion, the author should be commended because the manuscript is successful in linking the available data on exercise training in the elderly with the potentialities of this complex combination between conditioning and post-conditioning exercise. Nevertheless, I believe that the quality of the manuscript can be further improved based on some of the below-specified comments.

Major comments

  • In section #2 the authors discuss the physiology of post-activation performance enhancement (PAPE). It seems that the text dismisses that PAPE is different from post-activation potentiation (PAP) as it links the former with a physiological mechanism that is more associated with the latter (please, see Front Physiol. 2019; 10: 1359. For further details). The authors should extend this section of the text to provide a better distinction between PAPE and PAP; in addition, the mechanisms underlying PAPE should be further clarified. This would help the reader understanding some subsequent aspects of the text (e.g. duration of the pause between the conditioning and post-conditioning exercise).

  • I understand that PAPE has not been previously examined in the elderly. However, there are some data on PAP in older adults. Since PAP is mechanistically linked with PAPE, the authors should also review the available literature on the interaction between PAP and senescence (include this information in section #2, please).

Minor concerns

  • Ln 45, pp.2: what do you mean by “groups”? several people exercising at the same time?
  • Ln 73, pp2: replace “didn’t” with “did not”.
  • Ln 77, pp2: replace “fibers” with “fiber”.
  • Ln 80-82, pp2: awkward sentence – rewrite, please.
  • Ln 89, pp2: delete “results”. Ln 90, replace “reaping” with a synonym. Ln 92: use “it is” instead of “it’s” and “exercise” instead of “train”. Ln 93: delete “training” and in Ln 94, replace “training” with “exercise sessions”.
  • Ln 99, pp3: RM was not previously defined in the text. Ln 102: “conditioning activity” should appear as “CA” – the acronym was already defined in the text.
  • Ln 108-112, pp3: specify the number of reps and sets involved in these prescriptions known to be effective.
  • Ln 108, pp3: format “m2” as superscript for number 2.
  • Ln 110, pp3: delete “devices”. Ln 111: use “exercise” instead of “exercises”.
  • Ln 116, pp3: remove the dot after “min”.
  • Ln 128, pp3: “fatiguing exercise” or “CA”?
  • Ln 136, pp3: “investigated” instead of “were investigating”. Ln 137: vertical jump?. Ln 138: reactive strength? Ln 139: use “They are based”, “stretch-shortening cycle”, delete “which (…) primary function”, please. Ln 140-141: replace “immediately (…) phase” with “then shortened with a minimal transition delay”. Ln 142: replace “a movement” with “contractions” and “generates” with “generate”. Ln 143: replace “the shortening (…) movements” with “simple shortening contractions”.
  • Ln 157, pp4: delete “a” from “a simple”. Use “as safe choices” instead of “a not…”.
  • Ln 161: replace “counterparts “ with “a non-ballistic alternative”. Ln 162: replace “requirement” with “need”, use “during the concentric”, delete “in order”.
  • Ln 165: use “increases the cross-sectional” instead of “results in an incrased”.
  • Ln 166: replace “during” with “in”.
  • Ln 168: delete “adults”.
  • Ln 181: what do you mean by “breathing exercise”?
  • Ln 186-187: provide examples for these exercises, please.
  • Ln 187: replace “there” with “it”.
  • Ln 189: replace “use of” with “the use of”.
  • In each table: specify the duration of each mesocycle, please. Also the frequency of exercise sessions/week and the recovery between sessions. If possible, specify the duration of the macrocycle additionally, as well as the duration of the transition phase. It is important to provide these recommendations because as it is, it is impossible to determine whether this prescription should be applied on a daily basis (for instance).
  • Table #2. RPE4 does not specify reps – it is used as a subjective measure of intensity. I recommend that the authors use a combination of duration and RPE (e.g. 1 min at RPE4) to specify the frequency – maximum number of reps performed at RPE4 for 1 min (for example). In addition, min are not reps – please, revise this table to specify the number of reps as indicated in the upper row.
  • Table 2 legend: “caution”? “Fere”? “velocity loss threshold” (this was not explained in the text and the reader does not know what this is)?
  • Ln 202, pp5: close the sentence at “separately”. Begin a new sentence with “Therefore”. Ln 204-210: very long sentence. Break this sentence into 2 or 3 sentences.
  • Finally, provide a final paragraph with the practical application of this text.

Author Response

Manuscript Title: Utilization of post-activation performance enhancement in elderly adults

Manuscript Number: JCM1162818

General comments  

This manuscript provides a literary rationale for the use of complex training in the elderly. The text is well written and is easy to follow. Overall, in my opinion, the author should be commended because the manuscript is successful in linking the available data on exercise training in the elderly with the potentialities of this complex combination between conditioning and post-conditioning exercise. Nevertheless, I believe that the quality of the manuscript can be further improved based on some of the below-specified comments.

Reply: I would like to thank the Reviewer for this favourable opinion and for appreciating my scientific effort. I’m grateful for the commitment and very valuable comments that helped me to improve the quality of the manuscript. I hope that the revised manuscript adequately addresses all the raised issues.

Major comments

  • In section #2 the authors discuss the physiology of post-activation performance enhancement (PAPE). It seems that the text dismisses that PAPE is different from post-activation potentiation (PAP) as it links the former with a physiological mechanism that is more associated with the latter (please, see Front Physiol. 2019; 10: 1359. For further details). The authors should extend this section of the text to provide a better distinction between PAPE and PAP; in addition, the mechanisms underlying PAPE should be further clarified. This would help the reader understanding some subsequent aspects of the text (e.g. duration of the pause between the conditioning and post-conditioning exercise).

 Reply: Thank you for pointing this out, a very valuable comment. I have changed this paragraph: “A muscular phenomenon known under the term post-activation potentiation (PAP), which enhances muscular performances as a result of the recent voluntary contractile history [11,12], has become the focus of many strength and conditioning training programs. The primary mechanism responsible for PAP is the phosphorylation of the myosin regulatory light chain that leads to an increase in calcium sensitivity of the actomyosin complex [11,13]. Except that, the occurrence of performance enhancement may be also related to the other linked mechanisms, such as an increase in muscle temperature, fiber water content, and muscle activation which are assigned to the alternative term PAPE [13,14]. Therefore, the role of PAP on sports performance has been widely debated and it has been suggested that when a performance-related PAP approach is used instead of a mechanistic one, the term PAPE should be used [13,14]. In the case of training practice, increased force and power output production are induced by a potentiation complex consisting of a conditioning activity (CA) (i.e., bench press), followed by an explosive activity with a similar movement structure (i.e., medicine ball chest throw) [12,15]. This approach is called complex training. Since generating high values of muscle power is a critical variable for athletes in many sports, it seems that with caution and common sense, some inspiration from these solutions when training the elderly can be drawn. It is clear that the training methods used in sport are often extremely demanding and require adaptation and simplification to enable their implementation by older people, which is not impossible.”

I understand that PAPE has not been previously examined in the elderly. However, there are some data on PAP in older adults. Since PAP is mechanistically linked with PAPE, the authors should also review the available literature on the interaction between PAP and senescence (include this information in section #2, please).

Reply: Thank you for drawing attention to that. Excellent comment. I have added the following sentences: “A study by Hicks et al. [23] showed that twitch potentiation was significantly greater in young than in elderly adults (241% vs 166%). Interestingly, a later study showed that age-related differences in PAP occurred immediately after conditioning MVC and disappeared after 1 min [25]. This reduced PAP capacity in elderly adults is likely associated with a change in excitation-contraction coupling mechanisms [26] and factors related to muscle fiber atrophy [27,28]. In addition, a reduced myosin regulatory light chain phosphorylation during the CA has been observed in atrophied muscles and may have contributed to the diminished PAP in elderly individuals [29,30]. Even though the PAP response is reduced, it still occurs, which means that the elderly can benefit from properly designed training. For example, Hicks et al. [23] showed that the magnitude of potentiation significantly increased after 12 weeks of resistance training in the elderly (166% vs. 192%).”

Minor concerns

Ln 45, pp.2: what do you mean by “groups”? several people exercising at the same time?

Ln 73, pp2: replace “didn’t” with “did not”.

Ln 77, pp2: replace “fibers” with “fiber”.

Reply: Done.

Ln 80-82, pp2: awkward sentence – rewrite, please.

Reply: The whole paragraph has been changed.

Ln 89, pp2: delete “results”. Ln 90, replace “reaping” with a synonym. Ln 92: use “it is” instead of “it’s” and “exercise” instead of “train”. Ln 93: delete “training” and in Ln 94, replace “training” with “exercise sessions”.

Reply: Thank you for this comment. Corrected.

Ln 99, pp3: RM was not previously defined in the text. Ln 102: “conditioning activity” should appear as “CA” – the acronym was already defined in the text.

Reply: Thank you for point this out. Corrected.

Ln 108-112, pp3: specify the number of reps and sets involved in these prescriptions known to be effective.

Reply: Thank you, I have added it.

Ln 108, pp3: format “m2” as superscript for number 2.

Ln 110, pp3: delete “devices”. Ln 111: use “exercise” instead of “exercises”.

Ln 116, pp3: remove the dot after “min”.

Reply: Thank you very much for this comment. Corrected

Ln 128, pp3: “fatiguing exercise” or “CA”?

Reply: Thank you for point this out. Corrected.

Ln 136, pp3: “investigated” instead of “were investigating”. Ln 137: vertical jump?. Ln 138: reactive strength?

Reply: Thank you for point this out. Corrected.

Ln 139: use “They are based”, “stretch-shortening cycle”, delete “which (…) primary function”, please. Ln 140-141: replace “immediately (…) phase” with “then shortened with a minimal transition delay”. Ln 142: replace “a movement” with “contractions” and “generates” with “generate”. Ln 143: replace “the shortening (…) movements” with “simple shortening contractions”.

Reply: Thank you very much. Everything corrected.

Ln 157, pp4: delete “a” from “a simple”. Use “as safe choices” instead of “a not…”.

Ln 161: replace “counterparts “ with “a non-ballistic alternative”. Ln 162: replace “requirement” with “need”, use “during the concentric”, delete “in order”.

Ln 165: use “increases the cross-sectional” instead of “results in an incrased”.

Ln 166: replace “during” with “in”.

Ln 168: delete “adults”.

Reply: Thank you very much. Everything corrected.

Ln 181: what do you mean by “breathing exercise”?

Reply: I have changed to respiratory training and provide an example. “The solution that would have the slightest disturbance to the occurrence of the PAPE effect may be respiratory training (breathing exercises; i.e., diaphragm breathing), which could potentially contribute to the improvement of cognitive and pulmonary functions [49].”

Ln 186-187: provide examples for these exercises, please.

Reply: Thank you for point this out. I have added the following sentences: “Then, this could be progressed by the introduction of suspension exercises (e.g., jump squats using suspension straps). If it is not possible to introduce those solutions, exercise in a partial weight-bearing position might be a good choice (e.g., wall plyometric push-ups or wall sprints), followed by the ballistic exercises (e.g., jump squats).”

Ln 187: replace “there” with “it”.

Ln 189: replace “use of” with “the use of”.

Reply: Done.

In each table: specify the duration of each mesocycle, please. Also the frequency of exercise sessions/week and the recovery between sessions. If possible, specify the duration of the macrocycle additionally, as well as the duration of the transition phase. It is important to provide these recommendations because as it is, it is impossible to determine whether this prescription should be applied on a daily basis (for instance).

Reply: Thank you for this valuable comment. I have added another table with an example of the periodization model.

Table #2. RPE4 does not specify reps – it is used as a subjective measure of intensity. I recommend that the authors use a combination of duration and RPE (e.g. 1 min at RPE4) to specify the frequency – maximum number of reps performed at RPE4 for 1 min (for example). In addition, min are not reps – please, revise this table to specify the number of reps as indicated in the upper row.

Reply: Thank you for this valuable comment. I have corrected the table, moved RPE to the Intensity / External Load row and provide duration or repetition ranges where appropriate.

Table 2 legend: “caution”? “Fere”? “velocity loss threshold” (this was not explained in the text and the reader does not know what this is)?

Reply: Thank you for pointing this out. I have corrected the references and changed to “velocity loss control” to be consistent with the text and briefly define in line 206 “e.g. until mean barbell velocity dropped to 90% of that reached in the first repetition”

Ln 202, pp5: close the sentence at “separately”. Begin a new sentence with “Therefore”. Ln 204-210: very long sentence. Break this sentence into 2 or 3 sentences.

Finally, provide a final paragraph with the practical application of this text.

Reply: Thank you for this comment. I rewrote the whole paragraph and added limitations and practical implications of this text.